# Insulin-like Growth Factor 2 (IGF-2) and Insulin-like Growth Factor Binding Protein 7 (IGFBP-7) Are Upregulated after Atypical Antipsychotics in Spanish Schizophrenia Patients

**DOI:** 10.3390/ijms23179591

**Published:** 2022-08-24

**Authors:** Carlos Fernández-Pereira, Maria Aránzazu Penedo, Tania Rivera-Baltanas, Rafael Fernández-Martínez, Saida Ortolano, José Manuel Olivares, Roberto Carlos Agís-Balboa

**Affiliations:** 1Translational Neuroscience Group, Galicia Sur Health Research Institute (IIS Galicia Sur), Área Sanitaria de Vigo-Hospital Álvaro Cunqueiro, SERGAS-UVIGO, CIBERSAM-ISCIII, 36213 Vigo, Spain; 2NeuroEpigenetics Lab, University Hospital Complex of Vigo, SERGAS-UVIGO, 36213 Vigo, Spain; 3Rare Disease and Pediatric Medicine Group, Galicia Sur Health Research Institute (IIS Galicia Sur), SERGAS-UVIGO, 36312 Vigo, Spain; 4Department of Psychiatry, Área Sanitaria de Vigo, 36312 Vigo, Spain; 5Movement Disorders Group, Health Research Institute of Santiago de Compostela (IDIS), Área Sanitaria de Santiago de Compostela-Hospital Clínico Universitario de Santiago (CHUS), SERGAS-USC, 15706 Santiago de Compostela, Spain

**Keywords:** insulin-like growth factor (IGF-2), insulin-like growth factor binding protein 7 (IGFBP-7), schizophrenia (SZ), first episode (FE), drug-naïve

## Abstract

Insulin-like growth factor 2 (IGF-2) and IGF binding protein 7 (IGFBP-7) have been related to schizophrenia (SZ) due to their implication in neurodevelopment. The purpose of this study was to assess whether the alterations in IGF-2 and IGFBP-7 in SZ patients are intrinsically related to the psychiatric disorder itself or are a secondary phenomenon due to antipsychotic treatment. In order to test this hypothesis, we measured plasma IGF-2 and IGFBP-7 in drug-naïve first episode (FE) and multiple episodes or chronic (ME) SZ Caucasian patients who have been following treatment for years. A total of 55 SZ patients (FE = 15, ME = 40) and 45 healthy controls were recruited. The Positive and Negative Syndrome Scale (PANSS) and the Self-Assessment Anhedonia Scale (SAAS) were employed to check schizophrenic symptomatology and anhedonia, respectively. Plasma IGF-2 and IGFBP-7 levels were measured by Enzyme-Linked Immunosorbent Assay (ELISA). The FE SZ patients had much lower IGF-2, but not IGFBP-7, than controls. Moreover, both IGF-2 and IGFBP-7 significantly increased after atypical antipsychotic treatment (aripiprazole, olanzapine, or risperidone) in these patients. On the other hand, chronic patients showed higher levels of both proteins when compared to controls. Our study suggests that circulatory IGF-2 and IGFBP-7 increase after antipsychotic treatment, regardless of long-term conditions and being lower in drug-naïve FE patients.

## 1. Introduction

Schizophrenia (SZ) is a heterogeneous chronic psychiatric disorder that affects about 1% of the worldwide population, regardless of different countries, cultural groups, or gender [1]. SZ is characterized by positive symptoms (hallucinations, paranoid delusions, and disorganized speech), negative symptoms (decreased motivation and diminished emotional expression along with impaired social interaction), and general cognitive deficits involving executive functions, memory, and speed of mental processing [2]. The etiology of SZ is multifactorial and reflects a complex interaction between genetic variability and environmental contributors [3]. Nowadays, the clinical management of SZ is exclusively supported by subjective tools such as the Diagnostic and Statistical Manual of Mental Disorders V (DSM-V) for diagnosis or the Positive and Negative Syndrome Scale (PANSS) for evaluating either SZ severity or therapeutic efficacy as a percentage of reduction [4]. Therefore, research on new, objective biological markers based on non-invasive methods (e.g., plasma protein levels) is of the utmost importance for helping clinicians in early diagnosis and/or evaluating remission. Overall, it may help to understand the biological process underlying the progression of SZ [5]. These achievements could be implemented along with clinical interviews and psychiatric scales in order to define a more specific approach to different patients in an economical and safe manner [6].

Patients diagnosed with SZ debut with a first psychotic episode [7], and it is precisely in this onset period that it is extremely important to treat SZ as soon as possible to reduce adverse implications for long-term treatment outcomes [8]. Therefore, the development of tools that can help to define the physiological state of first episodes (FE) and discriminate them from multiple episodes (ME) (or chronic patients) should be considered priceless in clinical management [9].

Actually, there is a theory that suggests that SZ might arise as a result of an impaired neurodevelopmental process during early life [10]. In line with this, it is well-established that the Insulin-like growth factor (IGF) signaling system plays important roles in normal physiology [11] and brain development specifically [12], having a particular regulatory role in nerve growth survival, maturation, and proliferation in different brain cells such as astrocytes and oligodendrocytes [13]. The IGF system is mainly composed of three compartments: ligands (IGF-1, IGF-2), transmembrane receptors (IGF-1R, IGF-2R), and the IGF binding proteins (IGFBP-1 to 7) [14]. IGFs binding to the IGF-1R triggers the insulin receptor substrate (IRS)-initiated phosphatidylinositol 3-kinase (PI3K)–AKT/mammalian target of rapamycin (mTOR) pathway and the SHC-initiated Ras-mitogen-activated protein kinase (MAPK) pathway [15], which finally leads to an increase in cell proliferation, protein synthesis, and glucose metabolism, as well as a reduction in apoptosis [16].

On the other hand, IGFBPs act as serum carriers and passive inhibitors of IGF actions, limiting their bioavailability. However, IGFBPs also have IGF-independent functions [17]. IGFBP-3 is the most abundant IGFBP in postnatal blood and transports IGFs throughout the body, and also prevents IGFs from being degraded [18]. Conversely, IGFBP-7 exhibits a high binding affinity for insulin but low for IGFs [19], which makes serum IGFBP-7 levels an interesting target for studying type II diabetes mellitus [20,21]. Interestingly, the comorbidity between diabetes and SZ has been well-documented and attributed to some extent to lifestyle habits and antipsychotic therapy [22].

Moreover, other clinical blood variables such as lipid content, total cholesterol (TC), and triglycerides (TG) have received considerable attention in SZ since low levels of TC have been related to an increased risk of suicide attempts in FE patients [23]. What is more, TC and TG serum levels increase with antipsychotic treatment in responder patients but not in patients who had not met remission criteria [24]. Nonetheless, this improvement seemed to be dependent on drug type since patients who were following olanzapine monotherapy but not risperidone experienced an increase in TG that correlated with the reduction of PANSS total scores [25].

In the last decades, researchers have investigated the role of circulating IGF-1 protein levels in SZ [26,27,28,29,30,31,32,33,34,35]. Some studies found significantly higher levels of IGF-1 in a FE drug-naïve group of patients compared to healthy controls [26,27]. These levels were reduced after 10 weeks of risperidone monotherapy [27]. Nevertheless, one study had previously found that antipsychotic-naïve SZ patients had significantly lower levels of IGF-1 [28], while another measured no significant difference at baseline, but IGF-1 increased beyond healthy controls after 1 month of treatment [29]. In the case of SZ chronic patients, the majority of studies found that IGF-1 levels were statistically undifferentiated from healthy matched controls [30,31,32,33], and only one found that IGF-1 was downregulated in the SZ group [34]. Moreover, this downregulation was also observed in a responder group of SZ patients when compared not only against non-responders but also to healthy controls [35]. The nature of these contradictory results, as well as the clear relationship between the IGF signaling system and SZ, illustrates the importance of discriminating not only by stage, first vs. multiple, but also by responsiveness to treatment.

Despite the fact that IGF-2 has been less explored than IGF-1, circulating levels of IGF-2 have also met with some controversy. Initially, Akanji et al. (2007) showed that IGF-2 was upregulated in a chronic cohort of Arab SZ patients who were under stable antipsychotic medication, essentially haloperidol at a variable dose, which is not among the so-called group of “atypical” antipsychotic drugs [30]. Then, Yang et al. (2020) observed that IGF-2 was actually downregulated in a Chinese population of SZ patients that had been without treatment 3 months before protein measurement took place [36]. Shortly after, Chao et al. (2020) proved that IGF-2 was also downregulated in Chinese SZ patients, and interestingly, IGF-2 levels increased after 8 weeks of treatment regardless of the atypical antipsychotic administered (aripiprazole, clozapine, olanzapine, or risperidone) [37]. It is notable that the only study that has addressed IGF-2 gene expression found that the IGF-2 gene was the top downregulated gene in the prefrontal cortex of chronic SZ patients [38], which might represent the state of IGF-2 in circulation. However, IGFs are mainly synthesized in the liver in response to growth hormone (GH) signaling [39], and it has been proven in a rat model that IGFs are able to cross the blood–brain barrier and finally reach the brain [40]. Nonetheless, it is not clear whether circulating IGF-2 determines fluctuations in the brain [36].

IGFBPs have also been related to SZ. In particular, it was proven that measured plasma IGFBP-1 levels did not change after clozapine treatment [32], neither IGFBP-2 in patients that were under monotherapy either with olanzapine or risperidone [41]. Interestingly, the same tendency was observed in the study of Akanji et al. (2007) for IGFBP-3 [30], which again contradicts the downregulated IGFBP-3 levels found in Yang et al. (2020) [36]. It has to be mentioned that to the best of our knowledge, circulating levels of IGFBP-4, IGFBP-5, and IGFBP-6 have not yet been explored in SZ, although IGFBP-4 mRNA expression levels have been found downregulated in post-mortem hippocampal SZ brain sections [42]. Finally, IGFBP-7 circulating levels were found to be downregulated by Yang et al. (2020) [36].

The nature of these dissimilar results triggers the question of the true role of the IGF signaling system as a potential circulating biomarker in SZ. In this context, Yang et al. (2020) suggested that ethnicity, drug type, treatment regime, and SZ intrinsic heterogenicity might be the main reasons that could partially explain the contradictory nature of these results [36].

However, one of the main goals in mental disorders is to find a biomarker or a subset of them that can recreate changes in the subjective scales, such as PANSS in the case of SZ. In fact, both IGFs and IGFBPs have been previously researched for correlation with PANSS subscales in order to identify an objective biological signature that could represent SZ severity. Specifically, IGF-2 has been negatively correlated with the PANSS negative scale (PANSS-N), which might suggest that the increase in circulating IGF-2 could be a reflection of improvement in the negative symptoms in chronic SZ patients. Nevertheless, this was not the case for IGFBP-3 nor IGFBP-7 [36]. In the case of IGF-1, some studies found a positive correlation between PANSS-N and PANSS general (PANSS-G) [35], which may be pointing in the opposite direction. Others found a negative correlation between IGF-1 and PANSS-G and PANSS total scores (PANSS-T) [31], while another found no significant associations between IGF-1 and any of the PANSS subscales [26]. Intriguingly, no study has found a relationship between a protein belonging to the IGF signaling system and the PANSS positive subscale (PANSS-P), which actually represents the most characteristic externalization of a psychotic episode [43]. Alternatively, one study observed a negative correlation between IGF-1 and a test that evaluates hallucinations [28], which can be considered closely related to PANSS-P. Other scales, such as the Clinical Global Impression (CGI) and the Brief Psychiatric Rating Scale (BPRS), have been previously used in an effort to correlate IGF changes with SZ behavioral outcomes [35]. In our case, we decided to add the Self-Assessment Anhedonia Scale (SAAS) [44], which has been used before in different studies to characterize anhedonia from a physical, mental, and emotional perspective [45,46].

Members of our group have previously investigated the role of IGF-2 and IGFBP-7 in the extinction of fear memories [47,48] and as a potential drug target in Alzheimer’s disease (AD) [49]. Having in mind the controversy regarding IGFs and IGFBPs circulating levels in SZ, we investigated IGF-2 and IGFBP-7 plasma protein levels using a non-invasive method in a Spanish cohort of SZ patients, discriminating by FE and ME as was recently suggested by Yang et al. (2020) [36], and dividing the chronic group (ME) into responders and non-responders to treatment, according to expert clinical judgment. We put special emphasis on discerning how atypical antipsychotics (aripiprazole, olanzapine, or risperidone) altered IGF-2 and IGFBP-7 in FE drug-naïve patients and whether fluctuations in these proteins could reflect PANSS changes.

## 2. Results

### 2.1. Demographic and Clinical Data

Demographic and clinical data for all the groups considered are reported in Table 1. The 3 groups (C, FE, and ME) were not significantly different in terms of gender (χ^2^ (2, N = 99) = 2.37, *p* = 0.306), nor in median age (F (2) = 3.92, *p* = 0.141). There were no statistical differences among ME subgroups (ME_R vs. ME_NR and ME_T vs. ME_A) for gender (χ^2^ (1, *n* = 40) = 0.63, *p* = 0.427 and χ^2^ (1, *n* = 40) = 1.08, *p* = 0.298, respectively) and age (*t* (38) = 0.08, *p* = 0.937 and *t* (38) = 1.01, *p* = 0.32, respectively).

Interestingly, we found a positive overall correlation between IGF-2 and IGFBP-7 (r_s_ (99) = 0.562, *p* < 0.001). Moreover, this correlation was maintained in the control group (r_p_ (44) = 0.514, *p* < 0.001), and in the ME group (r_p_ (40) = 0.371, *p* = 0.018), but lost in the FE group (r_p_ (15) = 0.002, *p* =0.993).

Overall IGF-2 levels in males (*n* = 62, M = 116.20, SD = 53.98) did not follow a normal distribution (S-W (62) = 0.957, *p* = 0.03). Conversely, IGF-2 levels in females (*n* = 37, M = 124.61, SD = 46.55) did follow a normal distribution (S-W (37) = 0.986, *p* = 0.911) and so differences for IGF-2 between men and women were calculated with a *k* median comparison non-parametric test (F (1) = 0.017, *p* = 0.896) and no statistical differences were found. On the contrary, IGFBP-7 levels in both males (M = 69.91, SD = 23.93) and female (M = 66.60, SD = 24.24) were normally distributed (S-W (62) = 0.981, *p* = 0.443 and S-W (36) = 0.965, *p* = 0.286, respectively) and positive for Levene’s test (F (97) = 0.01, *p* = 0.922) and no statistical differences were found (*t* (97) = 0.662, *p* = 0.509). Age was not correlated with IGF-2 levels (r_s_ (99) = 0.132, *p* = 0.175) or with IGFBP-7 (r_p_ (99) = 0.150, *p* = 0.118) and so we did not include them in posterior analysis as covariates.

### 2.2. The Levels of IGF-2 and IGFBP-7 in Schizophrenia Patients and Healthy Controls

An analysis of variance (ANOVA) on IGF-2 and IGFBP-7 yielded significant variation among C, FE, and ME (F (2, 96) = 17.19, *p* < 0.001 and (2, 96) = 7.40, *p* = 0.001, respectively). We then applied Tukey’s multiple comparison post hoc test and found that the levels of IGF-2 were downregulated in the FE group when compared to the control group (Figure 1a, *p* = 0.0017). Additionally, both IGF-2 and IGFBP-7 protein levels were downregulated in the FE group when compared to the ME group (Figure 1a,b, *p* = 2.9 × 10^−7^ and *p* = 0.0017, respectively). The levels of both IGF-2 and IGFBP-7 were upregulated in the SZ chronic group of patients (ME) when compared to the control group (Figure 1a,b, *p* = 0.0065 and *p* = 0.0214, respectively).

Interestingly, a Wilcoxon paired test revealed that IGF-2 and IGFBP-7 circulating levels were significantly increased in the FE drug-naïve group after treatment (Figure 2a,b); the upregulation was more accentuated in IGF-2 (4.48 (8), *p* = 0.0078) than in IGFBP-7 (2.66 (10), *p* = 0.0137).

Intriguingly, an ANOVA analysis found significant differences in the levels of both IGF-2 (F (2, 81) = 4.84, *p* = 0.01) and IGFBP-7 (F (2, 81) = 4.23, *p* = 0.018) between ME_R, ME_NR and C. A post hoc Tukey’s multiple comparison analysis revealed that both IGF-2 and IGFBP-7 circulating levels were higher in the ME_R when compared to the control group (Figure 2c,d, *p* = 0.0192 and *p* = 0.0185, respectively). Conversely, we did not find any statistical difference in the levels of both proteins between the ME_R and ME_NR groups (Figure 2c,d, *p* = 0.4378 and *p* = 0.2350, respectively). Curiously, ANOVA shows significant differences between levels of both proteins among ME_A, ME_T and FE (IGF2: F (2, 52) = 28.80, *p* < 0.001 and IGFBP-7: F (2, 52) = 7.94, *p* = 0.001). Moreover, the FE group showed lower levels when compared to chronic patients that had either abandoned treatment (ME_A) or not (ME_T) before the episode took place (Figure 3a,b). However, this seemed not to be due to treatment abandonment as no statistical difference was found between ME_A and ME_T in IGF-2 or IGFBP-7 (Figure 3a,b, *p* = 0.9866 and *p* = 0.9530, respectively).

### 2.3. Correlation between IGF-2 and IGFBP-7 Plasma Protein Levels with PANSS and SAAS Scales

We did not find any significant correlation between IGF-2 levels and any of the PANSS subscales in all of the groups considered by the time an episode took place (Table 2). However, we did find positive correlations (Table 2) in the FE group after treatment (FE_1) with PANSS-P (r_p_ (8) = 0.745, *p* = 0.021), PANSS-N (r_p_ (8) = 0.862, *p* = 0.003), and PANSS-T (r_p_ (8) = 0767, *p* = 0.016), but not with PANSS-G (r_p_ (8) = 0.445, *p* = 0.230). On the other hand, IGFBP-7 positively correlated with the PANSS-N (Table 3, r_p_ (10) = 0.322, *p* = 0.043) in the ME group, but not in the FE (Table 3, r_p_ (10) = 0.41, *p* = 0.145). No other correlation was found between IGFBP-7 with any of the PANSS subscales in the FE after treatment (FE_1; Table 3).

On the other hand, we observed a positive correlation (Table 2; r_s_ (8) = 0.8, *p* = 0.014) between the difference in IGF-2 levels before and after treatment in the FE group (∆FE was calculated as the paired IGF-2 subtraction between FE_1 and FE) and PANSS-P percentage reduction (%↓ was calculated as (FE_1–FE)/FE × 100 for each PANSS subscale). Moreover, we then looked for a linear regression analysis between both variables (Figure 4. β = 0.3256), and variations in IGF-2 levels explained a significant proportion of variance in PANSS-P percentage reduction (*R^2^* = 0.5344, F (1, 7) = 8.04, *p* = 0.025).

Additionally, IGFBP-7 was also negatively correlated with the SAAS scale in the ME group (Table 3, r_s_ (40) = 0.325, *p* = 0.047).

### 2.4. Correlation between IGF-2 and IGFBP-7 Plasma Protein Levels and Metabolic Parameters

The levels of total serum cholesterol (TC) and triglycerides (TG) (Table 1) were found downregulated in the FE group when compared against the ME group (*t* (37) = 2.587, *p* = 0.0138 and *t* (37) = 2.078, *p* = 0.0447, respectively).

What is more, IGF-2 levels negatively correlated with TC in the FE group (Table 2, r_p_ (37) = −0.674, *p* = 0.016), but not in the ME group (Table 2, r_p_ (37) = 0.054, *p* = 0.788), neither in the case of IGFBP-7 (Table 3, r_p_ (37) = 0.038, *p* = 0.849).

The mean levels of glucose (F (5, 98) = 0.30, *p* = 0.91) and albumin (F (5, 84) = 0.54, *p* = 0.746) were not statistically different between SZ groups and no correlation was found between glucose and albumin with IGF-2 (Table 2) or IGFBP-7 (Table 3) in all of the SZ groups considered.

## 3. Discussion

The novelty of this study is that IGFBP-7 plasma protein levels have been evaluated in a cohort of FE drug-naïve SZ group of patients in Spain. To begin with, it caught our attention that IGFBP-7 and IGF-2 levels have never been questioned as possible correlated parameters in previous SZ studies [36]. Curiously, we found a positive correlation between IGF-2 and IGFBP-7 both in the healthy control and in the ME group, but interestingly, this correlation was completely lost in the FE group, which may be a distinguishing trait despite the fact that IGFBP-7 shows a weak binding affinity for IGF-2 in contrast with other IGFBPs [19]. For example, Akanji et al. (2007) found a positive strong correlation between IGF-2 and IGFBP-3 (r_s_ = 0.82; *p* < 0.01) [30].

As has been previously mentioned, IGF-2 has been found to be either upregulated [30] or downregulated [36,37]. In our case, we have found that the levels of both IGF-2 and IGFBP-7 are upregulated in chronic patients when compared to healthy controls, and so far, it constitutes the first data on a Spanish (Caucasian) SZ cohort. However, when we decomposed the ME group, we found that the responder (ME_R) but not the non-responder (ME_NR) group has both IGF-2 and IGFBP-7 upregulated compared to healthy controls. Yang et al. (2020) reason that these results may suggest the importance of including treatment responsiveness as an important point in SZ heterogenicity when studying circulating IGFs or IGFBPs protein levels in SZ [36].

On the other hand, the levels of IGF-2, but not IGFBP-7, were significantly reduced at baseline in the FE when compared to controls. Levels of IGFBP-7 were reduced but did not reach statistical meaning. Additionally, both IGF-2 and IGFBP-7 were downregulated in the FE in comparison to the ME group, which could be interpreted as a result of long-term atypical antipsychotic treatment.

However, the levels of both IGF-2 and IGFBP-7 significantly increased in the FE drug-naïve group of patients after a short period with atypical antipsychotics (aripiprazole, olanzapine, or risperidone), suggesting that the increase could be mediated by the treatment itself regardless of long-term conditioning and even be drug-type independent, as was recently observed for IGF-2 in Chao et al.’s (2020) study [37].

The fact that chronic patients who had abandoned treatment showed similar IGF-2 and IFGBP-7 levels as patients who were actually following treatment before the episode took place might shed some light in the opposite direction. Therefore, the levels of these proteins might not rapidly fluctuate in relation to treatment, at least from the perspective of chronic patients who have been undergoing psychotherapy for years. On average, our patients abandoned treatment at least 1 month before the episode took place. In the case of Yang et al.’s (2020) study, their patients were without treatment 3 months before protein measurement, and they found that both IGF-2 and IGFBP-7 were downregulated compared to controls [36]. Taken together, these results could imply that between 1 to 3 months is required to reset IGF levels to FE drug-naïve baseline levels.

However, it has been postulated that IGF-2 circulating levels could depend on ethnicity, being higher in the plasma of Caucasians than in Asian individuals [50]. As Yang et al. (2020) [36] already explained, the ethnic approach could be the reason behind the differences observed in their study (downregulated) and in both our work and Akanji et al.’s (2007) [30] research (upregulated), since three different ethnic groups were considered: Asian, Caucasian, and Arab, respectively. From our perspective, ethnicity could explain absolute differences in IGFs or IGFBPs circulating values in healthy individuals, but change itself, if it is considered as being a result of a mental disorder or even another specific confounding factor, should be consistently pointing in the same direction (either upregulated or downregulated), at least if we seek to use it as a reliable global biomarker. However, this is still a matter of debate, and more studies with bigger sample sizes are needed to answer this question.

In spite of the above-mentioned controversy, IGF-2 downregulation has been proposed as a susceptibility factor in multiple mental disorders such as depression, bipolar disorder, anxiety, and even Alzheimer’s disease (AD) [51]. The theoretical way IGF-2 deficiency disrupts healthy conditions would be through memory impairment and by altering cognitive processes. Nevertheless, this hypothesis has also met some contradictory results. One study measured that IGF-2 serum levels were upregulated [52], and in another, these levels were found undifferentiated [53] in AD patients when compared against matched-age healthy controls. Curiously, as far as we have found, no study has yet evaluated either IGF-2 or IGFBP-7 circulating levels in patients diagnosed with bipolar disorder or depression. This could be a suitable premise for future research since depression [54], bipolar disorder [55], and other psychiatric or mood alterations such as post-traumatic stress or anxiety disorder [56,57] are considered important risk factors for AD.

From our perspective, one of the biggest interests when finding biomarkers is that they can recreate changes in subjective scales that actually have clinical use. The PANSS scale is one of the gold standards for evaluating SZ severity.

In our case, we found one positive correlation between PANSS-N and IGFBP-7 by the time the episode took place in the ME group. Therefore, higher levels of this protein at that specific point might be reflecting severe PANSS-N punctuations. On the other hand, IGF-2 did not correlate with any of the PANSS subscales by the time the episode erupted. Nonetheless, the FE group after treatment positively correlated with PANSS-P, PANSS-N, and PANSS-T, but not with PANSS-G. These results contrast with the negative correlation that was found in Yang et al. (2020) [36]. Nevertheless, that correlation has not been observed in other studies, but they have noticed a significant correlation between the increase in IGF-2 levels and the reduction of PANSS-N scores in patients that had been through therapy with atypical antipsychotics [37]. IGFBP-7 has been less studied in this context, and to our knowledge, no correlation has been found with any PANSS subscale, and the same was observed for IGFBP-3 [36].

Finally, we found a correlation between the variation of IGF-2 circulating levels with PANSS-P percentage reduction. However, we did not find any correlation with the other two subscales (PANSS-N and PANSS-G) or with the PANSS total score. The positive symptoms are related to hallucinations and delirium, and, usually, they are the main target of antipsychotic drugs, whereas the cognitive and negative symptoms are more passive and tend to be more constant and less affected by treatment [58]. Somehow, finding a robust correlation between a protein or a subset of proteins and positive symptoms could allow anticipating the occurrence of an episode and acting as a predictive or diagnostic biomarker [59]. To evaluate negative symptoms, the SAAS scale could be of help since it measures three main domains of anhedonia (physical, emotional, and intellectual) [44]. In our case, anhedonia tended to increase with the course of the SZ. Even more, we found a positive correlation between IGFBP-7 and the SAAS scale in the ME group. The fact that we found a similar correlation between the PANSS-N and the SAAS scale with IGFBP-7 may be because both scales describe the same phenomena; that is, the negative symptoms and/or anhedonia generally correlate with IGFBP-7 levels.

Serum lipid deregulations have been closely related to mental disorders [23,24,25]. In the case of SZ, lower levels of TC have been postulated to increase the risk of suicide attempts in FE patients [23]. Additionally, it has been observed that TC and TG serum levels increase with antipsychotic treatment in ME patients who responded to treatment [24]. This is partly in accordance with our results since FE patients showed lower TC and TG levels when compared to ME, but we did not find higher levels in the ME_R group when compared to the ME_NR group of SZ patients. What is more, Akanji et al. (2007) found a positive correlation between IGF-2 and TC levels in SZ patients [30]. Conversely, we found a negative correlation in the FE between IGF-2 and TC levels. In the case of TG levels, no correlation with IGF-2 has been found either in previous studies or in our current research.

### Limitations and Future Perspectives

This study has the following limitations. First, despite the fact that we found statistical differences between controls and patients, the mechanism through which the IGF signaling system affects schizophrenia is still unknown. Future research in animal models could shed some light on IGF-2 signaling in SZ-related behavior patterns. Second, whether peripheral proteins represent changes in the central nervous system was not assessed. Further experiments analyzing cerebral spinal fluid in human samples might be an interesting approach. Third, although we included albumin content and other metabolic-related parameters such as glucose, cholesterol, and triglycerides, other variables, such as body mass index, could have been of interest in searching for confounding factors. Fourth, the IGF signaling system might be affected by cognitive status and should have been assessed. For future studies, we recommend including at least one test that assesses cognitive status. Fifth, we could not measure glucose, TC, TG, and albumin in FE patients after treatment, which might have been interesting in order to find if their levels would have changed in response to antipsychotic therapy. Finally, a few FE patients could not finish the entire study due to medical complications, so we had to reduce the sample size in the statistical paired analysis. Additionally, having a complete panorama of the IGF signaling system status might be considered in SZ and other psychiatric disorders, such as major depression disorder, due to its implication for cognition.

## 4. Materials and Methods

### 4.1. Experimental Design

Here we present a cross-sectional observational study that started in September 2018 and ended in November 2021. For this study, we recruited 55 schizophrenic patients (*n* = 55) who met the DSM-V diagnostic criteria at Álvaro Cunqueiro Hospital (Vigo, Spain) between 2018 and 2021. Additionally, 44 healthy individuals were donors and included as a control group (*n* = 45).

Inclusion criteria included meeting the DSM-V schizophrenic diagnostic criteria, age ≥18 years old, and delivery of signed written consent. The exclusion criteria included additional DSM-V diagnosis, medical, neurologic pathologies, or other medical conditions such as pregnancy or lactation. Patients recruited for this study were under different stages of the psychiatric disorder and categorized as first episode drug-naïve (FE; *n* = 15) and multiple episode or chronic patients (ME; *n* = 40). The ME group was divided into treatment responders (ME_R; *n* = 19) and non-responders (ME_NR; *n* = 21) to previous treatment.

FE episode patients underwent treatment with atypical antipsychotics (aripiprazole, olanzapine, or risperidone) after blood collection, with dosages depending on the judgment of the clinician. We called this group FE_1 (after treatment). Some patients from the FE group decided not to end the study.

All patients and controls that participated in this study were of Spanish nationality. We carried out this research according to the requirements of the Declaration of Helsinki, and we obtained the corresponding ethical approval. We also received written consent from all patients or their corresponding legal guardians.

### 4.2. Blood Collection

The blood collection took place in the morning between 8 and 10 a.m. after an overnight fasting period in EDTA tubes. Plasma was immediately separated with Ficoll–Paque (2.5 mL) by centrifugation (2000 rpm, 35 min) and then aliquoted and stored in the freezer (−80 °C) until protein measurement.

### 4.3. Measurement of Psychopathological Symptoms (PANSS) and Anhedonia (SAAS)

In our study, a neuropsychologist belonging to the Álvaro Cunqueiro Hospital was responsible for measuring the psychopathological symptoms of all patients by using the Positive and Negative Syndrome Scale (PANSS), which is divided into three categories, each one encompassing a different mark: PANSS positive (PANSSP, 7–49 points), PANSS negative (PANSSN, 7–49), PANSS general (PANSSG, 16–122), and PANSS total (PANSST, 30–220), which is the sum of the other three. The response to treatment was checked by an interview with an expert psychiatrist.

We used the Self-Assessment Anhedonia Scale (SAAS) created by Olivares, Berrios, and Bousoño in 2005 [44] to measure anhedonia among schizophrenic patients and controls. The scale measures 3 different domains of anhedonia (physical, social, and intellectual) and is composed of 27 items, each with 3 possible answers (intensity, frequency, and change) on a Likert-like scale from 0 to 10. The total punctuation in each subscale ranges from 0 to 270 points, which makes a total of 810 points.

### 4.4. Metabolic Parameters Measurement

Clinicians examined blood samples from SZ patients in a regular blood test to measure serum total cholesterol levels (TC; mg/dL), total triglycerides (TG; mg/dL), glucose (mg/dL), and total albumin content (g/dL) by the time patients were hospitalized.

### 4.5. Measurement of IGF-2 and IGFBP-7 Plasma Proteins

We measured the levels of IGF-2 (Catalog N° EH0166) and IGFBP-7 (Catalog N° EH01710) plasma proteins using an ELISA commercially available kit (Wuhan, China. Fine Biotech Co., Ltd.). First, the kits were tested for three dilution factors (1/50, 1/100, and 1/200) as it is recommended by the fabricant.

Briefly, equal volumes of the standard or samples were added to the corresponding microplate wells with a biotin-conjugated antibody specific to IGF-2 or IGFBP-7. Next, avidin conjugated to horseradish peroxidase (HRP) was added to each well and incubated. Thereafter, TMB substrate solution was added, and only the wells that contained either IGF-2 or IGFBP-7 exhibited a change in color from transparent to a gradient-scaling blue. After incubation at 37 °C, the colorimetric reaction was terminated with a so-called stop solution (sulfuric acid), and the color finally changed from blue to a degree of yellows. Then, the microplates were measured spectrophotometrically at a wavelength of 450 nm. In order to calculate protein concentration, we subtracted the zero standard OD and then constructed a standard hyperbolic curve by plotting the mean OD obtained and known standard concentrations. The levels of IGF-2 and IGFBP-7 plasma proteins in the samples were then obtained by comparing the OD of the sample to the standard curve.

### 4.6. Statistical Analysis

Quantitative data are reported as mean and standard deviation (M ± SD) for each parameter. We used the Shapiro–Wilk test to check whether quantitative variables (IGF-2, IGFBP-7, age, PANSS, SAAS, glucose, TC, TG, and albumin) were following a normal distribution (S-W (df) = F, *p* > 0.05) or not. If variables were normally distributed and positive for Levene’s test, we used a parametric Student’s *t*-test (t (df) = F, p) or one-way analysis of variance (ANOVA) (F (df1, df2) = F, p) for mean comparison between groups applying post hoc Tukey’s multiple comparison test. If not, we used a non-parametric test, and medians were compared instead. For the assessment of the relationships between quantitative variables, we used either Pearson’s correlation coefficient (r_p_ (df) = r_p_, p) if both variables were following a normal distribution and Spearman’s (r_s_ (df) = r_s_, p) if not. If a quantitative variable showed a statistical correlation with either IGF-2 or IGFBP-7 (*p* < 0.05), we then adjusted for that variable with analysis of covariance (ANCOVA) (F (df1, df2) = F, p) to determine the influence of those factors in the comparison between groups. We also performed a Wilcoxon’s test (W (df) = W, p) for the comparison of IGF-2 and IGFBP-7 paired samples in the FE group before and after treatment (FE vs. FE_1).

We used G*Power 3.1.9.7 Software [60] to estimate the total sample size per group. We selected the F test family and ANOVA as statistical tests. We estimated an effect size of 0.5, significance of 95% (α = 0.05), and potency of 90% (β = 0.01). The number of groups was fixed at 3 (HC, FE, and ME). As a result, we obtained *n* = 91 per group. However, this estimation had to meet two main limitations. First, in the case of FE, we used incidence instead of prevalence (as we used for ME patients). Second, the COVID-19 pandemic made the recruitment process difficult because blood extraction occurred when these patients were hospitalized, as indicated in the blood collection (Materials and Methods) section.

## 5. Conclusions

In this study, we found that both IGF-2 and IGFBP-7 circulating levels tend to increase with atypical antipsychotic treatment in a cohort of Spanish schizophrenic patients. Presumably, we can conclude that this upregulation is not just due to long-term treatment conditioning since this increase was also measured in debutant schizophrenic patients who were drug-naïve. Moreover, several correlations between the IGF signaling system and schizophrenia positive and negative symptomatology were found. However, the role of the IGF signaling system in schizophrenia is still not clear, and ethnicity, drug type, treatment regimen, SZ intrinsic heterogenicity, and metabolic and cognition status might be reasons that could partially explain the nature of these results. Future research on how other IGF signaling members change with treatment in schizophrenia patients is needed, considering the above-mentioned aspects and with a larger sample size.

## Figures and Tables

**Figure 1 ijms-23-09591-f001:**
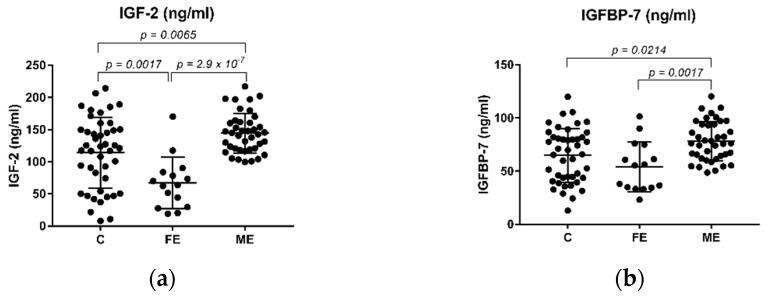
Plasma levels of IGF-2 (ng/mL) and IGFBP-7 (ng/mL) in SZ patients and healthy controls. (**a**) Levels of IGF-2 are significantly downregulated in the FE group when compared to the healthy control and the ME groups; (**b**) Levels of IGFBP-7 are significantly downregulated in the FE group compared to the ME group. Tukey’s multiple comparison test was used, and significant *p*-values are shown in each graph. FE: First Episode drug-naïve patients, ME: multiple or chronic patients.

**Figure 2 ijms-23-09591-f002:**
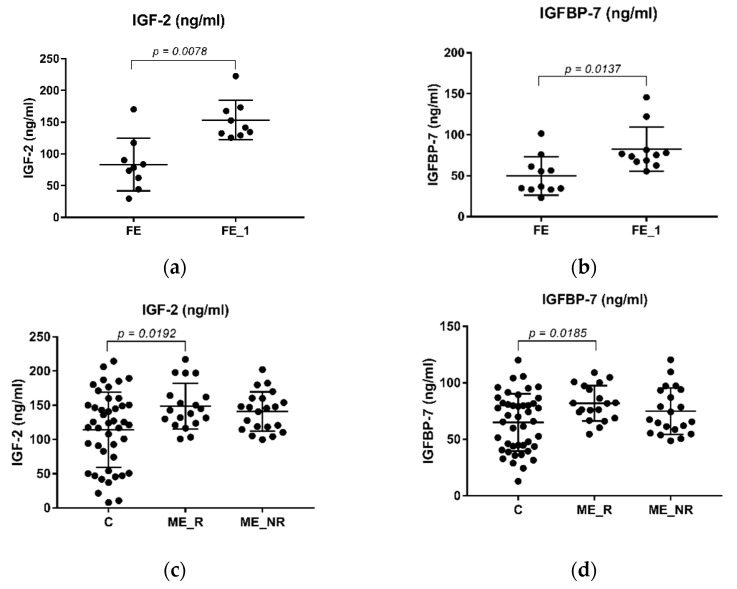
Plasma levels of IGF-2 (ng/mL) and IGFBP-7 (ng/mL) in SZ patients. (**a**) Levels of IGF-2 are significantly upregulated in related samples of FE_1. (**b**) Levels of IGFBP-7 are significantly upregulated in related samples of the FE_1 group. A Wilcoxon paired test was applied in both (**a**,**b**), and significant *p*-values are shown in each graph. No statistical difference was found in both (**c**) IGF-2 and (**d**) IGBFP-7 levels between ME_R and ME_NR. The levels of both IGF-2 (**c**) and IGFBP-7 (**d**) were found upregulated in the ME_R compared to the control group. An unpaired t-test was performed to calculate *p*-values. Significant *p*-values are given in each graph. FE: First Episode drug-naïve patients, FE_1: First Episode group after treatment, ME: multiple or chronic patients, ME_R: ME patients who had previously responded to treatment, ME_NR: ME patients who had not responded to treatment.

**Figure 3 ijms-23-09591-f003:**
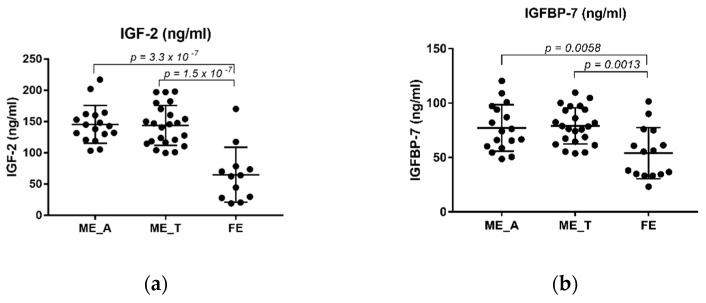
Plasma levels of IGF-2 (ng/mL) and IGFBP-7 (ng/mL) in SZ patients. (**a**) Levels of IGF-2 are significantly downregulated in the FE group when compared against both ME_A and ME_T; (**b**) Levels of IGFBP-7 are significantly downregulated in the FE group when compared against both ME_A and ME_T. Tukey’s multiple comparison test was applied, and significant *p*-values are shown in each graph. FE: First Episode drug-naïve patients, ME: multiple or chronic patients, ME_A: ME who had abandoned treatment at least 1 month before the psychotic episode took place, and ME_T: ME who had been following treatment until the psychotic episode took place.

**Figure 4 ijms-23-09591-f004:**
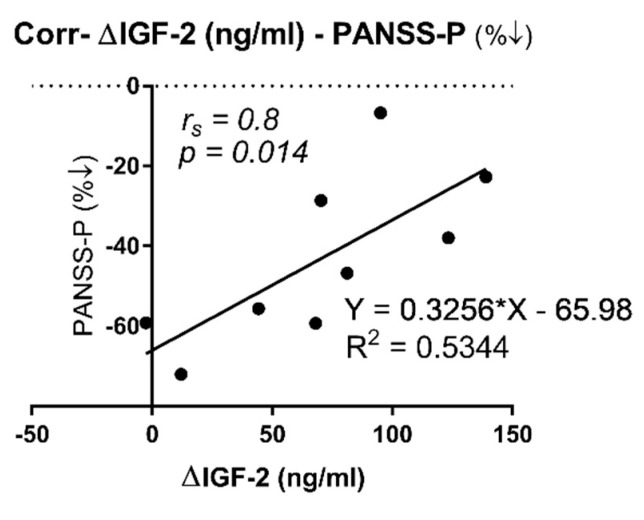
Correlation between ∆IGF-2 circulating levels and PANSS-P percentage reduction in first episode, drug-naïve patients before and after treatment with atypical antipsychotics (aripiprazole, olanzapine, and risperidone). ∆IGF-2 (ng/mL): paired protein difference between FE_1 and FE (∆FE = FE_1–FE), PANSS-P (%↓): indicates PANSS percentage reduction calculated as (FE_1–FE)/FE × 100. (r_s_) indicates Spearman’s correlation coefficient and *p*-value is given. Trend line calculated by linear regression analysis and the correspondent R-squared (R^2^) is shown.

**Table 1 ijms-23-09591-t001:** Demographic and clinical data.

Variable	Control (*n* = 45)	FE (*n* = 15)	ME (*n* = 40)	ME_R (*n* = 19)	ME_NR (*n* = 21)	ME_A (*n* = 16)	ME_T (*n* = 23)
		FE_1 (*n* = 11)					
Sex (F/M)	20/25	4/11	13/27	5/14	8/13	4/12	9/14
		3/8					
Age (years)	41.22 ± 10.58	31.73 ± 15.53	39.85 ± 11.26	40.00 ± 12.15	39.71 ± 10.69	37.69 ± 9.12	41.39 ± 12.73
		32.36 ± 16.24					
IGF-2 (ng/mL)	114.24 ± 55.00	66.93 ± 39.99	144.06 ± 30.84	148.64 ± 33.41	140.95 ± 28.64	144.68 ± 30.99	143.87 ± 31.93
		153.45 ± 30.98					
IGFBP-7 (ng/mL)	64.9174 ± 25.36	54.04 ± 23.49	78.28 ± 18.55	81.98 ± 15.61	74.93 ± 20.67	76.57 ± 21.90	79.09 ± 16.65
		82.47 ± 27.04					
PANSS-P	-	21.27 ± 5.02	22.92 ± 8.12	22.21 ± 8.16	23.57 ± 8.22	22.25 ± 9.73	22.91 ± 6.81
		11.33 ± 4.06					
PANSS-N	-	24.93 ± 8.92	27.42 ± 8.65	26.79 ± 7.44	28.00 ± 9.76	24.81 ± 6.92	29.04 ± 9.55
		16.78 ± 7.33					
PANSS-G	-	35.93 ± 9.38	38.15 ± 8.13	38.37 ± 7.65	37.95 ± 8.73	37.69 ± 9.16	38.30 ± 7.69
		23.22 ± 5.99					
PANSS-T	-	82.13 ± 19.37	88.5 ± 21.45	87.37 ± 19.20	89.52 ± 23.73	84. 75 ± 23.38	90.26 ± 20.31
		51.33 ± 15.66					
SAAS	108.17 ± 59.32	155.33 ± 60.54	198.55 ± 89.67	172.86 ± 94.93	181.50 ± 88.51	205.21 ± 85.42	196.48 ± 95.32
Glucose (mg/dL)	-	87.14 ± 22.68	85.57 ± 14.17	84.62 ± 14.14	86.64 ± 14.65	82.54 ± 7.98	88.56 ± 17.74
TC ^1^ (mg/dL)	-	142.75 ± 24.82	185.41 ± 54.35	170.93 ± 56.67	201.00 ± 49.13	184.31 ± 51.06	191.23 ± 58.67
TG ^2^ (mg/dL)	-	77.75 ± 31.64	116.11 ± 60.04	103.57 ± 49.04	129.61 ± 69.45	117.77 ± 62.18	118.62 ± 60.74
Albumin (g/dL)	-	4.1225 ± 0.26	4.1223 ± 0.44	4.22 ± 0.40	4.01 ± 0.48	4.04 ± 0.41	4.22 ± 0.47

The mean values and the standard deviation of the different variables considered are shown. FE: First Episode drug-naïve patients, FE_1: First Episode group after treatment, ME: multiple or chronic patients, ME_R: ME patients who had previously responded to treatment, ME_NR: ME patients who had previously not responded to treatment, ME_A: ME who had abandoned treatment at least 1 month before the last psychotic episode took place and ME_T: ME who had been following treatment until the last psychotic episode took place. Sample size (N) is indicated for each column. ^1^ TC: total cholesterol, ^2^ TG: total triglycerides.

**Table 2 ijms-23-09591-t002:** Correlation between IGF-2 and variables.

Protein/Group/Coefficient			Variables								
			PANSS-P	PANSS-N	PANSS-G	PANSS-T	SAAS	Glucose (mg/dL)	TC^1^ (mg/dL)	TG^2^ (mg/dL)	Albumin (g/dL)
**IGF-2 (ng/mL)**	**FE (*n* = 15)**	r	0.003	−0.314	−0.286	−0.282	−0.336	0.002	**−0.674**	0.003	−0.147
		*p*	0.991	0.255	0.301	0.308	0.241	0.993	**0.016 ***	0.992	0.648
	**FE_1 (*n* = 9)**	r	**0.745 ^s^**	**0.862 ^s^**	0.445 ^s^	**0.767 ^s^**	-	-	-	-	-
		*p*	**0.021 ***	**0.003 ***	0.230	**0.016 ***	-	-	-	-	-
	**∆FE (*n* = 9)**	r	**0.800 ^s,%↓^**	0.150 ^s, %↓^	0.117 ^s, %↓^	0.533 ^s, %↓^	-	-	-	-	-
		*p*	**0.014 ***	0.708	0.776	0.148	-	-	-	-	-
	**ME (*n* = 40)**	r	0.135 ^s^	0.162	−0.035	0.075	0.026 ^s^	0.08	0.054	0.142	0.160
		*p*	0.407	0.318	0.829	0.644	0.879	0.673	0.788	0.479	0.435
	**ME_R (*n* = 19)**	r	0.226	0.168	−0.090	0.125	−0.271	0.269	−0.275 ^s^	0.015	0.347
		*p*	0.353	0.491	0.714	0.609	0.277	0.314	0.342 ^s^	0.958	0.224
	**ME_NR (*n* = 21)**	r	−0.085	0.257 ^s^	0.008	0.049	0.259	−0.258 ^s^	0.450	0.284	−0.04
		*p*	0.715	0.261 ^s^	0.974	0.834	0.270	0.374 ^s^	0.123	0.347	0.891
	**ME_A (*n* = 16)**	r	−0.084 ^s^	−0.169	−0.273	−0.235	0.095 ^s^	−0.466 ^s^	0.138	0.110	0.201
		*p*	0.757 ^s^	0.530	0.307	0.381	0.748 ^s^	0.108 ^s^	0.653	0.720	0.491
	**ME_T (*n* = 23)**	r	0.267	0.256 ^s^	0.141	0.296	−0.008 ^s^	0.219	−0.007	0.207	0.094
		*p*	0.217	0.238 ^s^	0.522	0.170	0.971 ^s^	0.416	0.983	0.498	0.770

^s^ Spearman’s correlation coefficient, * *p*-value < 0.05. Coefficient values range from 1 to −1; the closer the coefficient to extreme values, the stronger the correlation between variables. A positive correlation indicates that both values vary in the same direction. A negative correlation indicates an inverse variation. Pearson’s correlation coefficient (r_p_) was employed when data followed a normal distribution (Shapiro–Wilk; *p* > 0.05). FE: First Episode drug-naïve patients, FE_1: First Episode group after treatment, ∆FE: Protein difference between FE_1 and FE (∆FE = FE_1–FE), ^%^^↓^: indicates PANSS percentage reduction calculated as (FE_1–FE)/FE × 100, ME: multiple or chronic patients, ME_R: ME patients who had previously responded to treatment, ME_NR: ME patients who had not responded to treatment, ME_A: ME who had abandoned treatment at least 1 month before the psychotic episode took place, and ME_T: ME who had been following treatment until the psychotic episode took place. ^1^ TC: total cholesterol, ^2^ TG: total triglycerides.

**Table 3 ijms-23-09591-t003:** Correlation between IGFBP-7 and variables.

Protein/Group/Coefficient			Variables								
			PANSS-P	PANSS-N	PANSS-G	PANSS-T	SAAS	Glucose (mg/dL)	TC^1^ (mg/dL)	TG^2^ (mg/dL)	Albumin (g/dL)
**IGFBP-7 (ng/mL)**	**FE** **(*n* = 15)**	r	−0.176	0.41	0.252	0.265	−0.03	−0.172	0.067	0.177	−0.079
		*p*	0.548	0.145	0.385	0.36	0.918	0.556	0.83	0.583	0.808
	**FE_1** **(*n* = 11)**	r	0.068	0.128	−0.119	−0.036	-	-	-	-	-
		*p*	0.841	0.709	0.727	0.915	-	-	-	-	-
	**∆FE** **(*n* = 11)**	r	0.136 ^s, %↓^	−0.209 ^s, %↓^	−0.200 ^s, %↓^	−0.100 ^s, %↓^	-	-	-	-	-
		*p*	0.694	0.539	0.557	0.776	-	-	-	-	-
	**ME** **(*n* = 40)**	r	0.081 ^s^	**0.322**	0.241	0.276	**0.325 ^s^**	−0.168	0.038	0.037	0.098
		*p*	0.62	**0.043 ***	0.135	0.085	**0.047 ***	0.374	0.849	0.855	0.633
	**ME_R** **(*n* = 19)**	r	0.048	0.137	0.123	0.123	0.021	−0.324	−0.288 ^s^	−0.301	−0.398
		*p*	0.845	0.576	0.615	0.617	0.933	0.221	0.318 ^s^	0.296	0.158
	**ME_NR** **(*n* = 21)**	r	0.246	0.435 ^s^	0.311	0.384	0.384	0.075 ^s^	0.263	0.185	0.441
		*p*	0.283	0.049 ^s^	0.170	0.086	0.094	0.799 ^s^	0.385	0.545	0.151
	**ME_A** **(*n* = 16)**	r	−0.013 ^s^	0.330	0.288	0.250	0.407 ^s^	−0.219 ^s^	−0.171	−0.182	0.049
		*p*	0.961 ^s^	0.212	0.279	0.349	0.149 ^s^	0.472 ^s^	0.577	0.552	0.869
	**ME_T** **(*n* = 23)**	r	0.172	0.316 ^s^	0.178	0.276	0.363 ^s^	−0.132	0.284	0.353	0.099
		*p*	0.433	0.142 ^s^	0.417	0.203	0.089 ^s^	0.625	0.347	0.236	0.76

^s^ Spearman’s correlation coefficient, * *p*-value < 0.05. Coefficient values range from 1 to −1; the closer the coefficient to extreme values, the stronger the correlation between variables. A positive correlation indicates that both values vary in the same direction. A negative correlation indicates an inverse variation. Pearson’s correlation coefficient (r_p_) was employed when data followed a normal distribution (Shapiro–Wilk; *p* > 0.05). FE: First Episode drug-naïve patients, FE_1: First Episode group after treatment, ∆FE: Protein difference between FE_1 and FE (∆FE = FE_1–FE), ^%^^↓^: indicates PANSS percentage reduction calculated as (FE_1–FE)/FE × 100, ME: multiple or chronic patients, ME_R: ME patients who had previously responded to treatment, ME_NR: ME patients who had not responded to treatment, ME_A: ME who had abandoned treatment at least 1 month before the psychotic episode took place, and ME_T: ME who had been following treatment until the psychotic episode took place. PANSS: Positive and Negative Syndrome Scale, SAAS: Self-Assessment Anhedonia Scale, ^1^ TC: total cholesterol, ^2^ TG: total triglycerides.

## Data Availability

Not applicable.

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
