# Peer review of "Insulin-like Growth Factor 2 (IGF-2) and Insulin-like Growth Factor Binding Protein 7 (IGFBP-7) Are Upregulated after Atypical Antipsychotics in Spanish Schizophrenia Patients"

_ijms, 2022, doi:10.3390/ijms23179591_

Round 1

Reviewer 1 Report

Dear Author,

Thanks for submitting your research manuscript entitled "Insulin-like growth factor 2 (IGF-2) and Insulin-like growth factor binding protein 7 (IGFBP-7) plasma protein levels are upregulated after atypical antipsychotic treatment in first episode drug-naïve schizophrenic patients ".

Before giving my final comments as well as the final revision of this manuscript, the author needs to address the following comments scientifically.
Major concerns:
Please find out the following comments

·         The rationale and purpose behind selecting the  Insulin-like growth factor (IGF) signaling system has been related with schizophrenia 16 (SZ) process are incomplete.

·         Title is misleading the reader. Title needs to reframed in simply manner accordingly.

·         The reviewer found irrational and non-scientific justification in the abstract—introduction and discussion part.

·         Abstract is very confusing. Irrational and fused with repetitions. Scientific output is not clear with this abstract.

·         The results and discussion are very poorly explained.

·         The reviewer feels the author needs to elaborate and justify it with proper citations and strong evidence. The author fails to explain the relevant justification in the introduction as mentioned in the discussion part.

·         A major drawback is a lack of supporting pre-clinical and clinical evidence regarding targeting drugs.

·         Complete mismatch of abstract, introduction, results and discussion.

·         Old and outdated references are not allowed. Try to update with relevant.

Title:

·         Mismatch of title with relevant introduction and conclusive remarks in the conclusion part.

Abstract:

-     The rationale behind this research is not well explained, and several major concerns still constrain the reviewer's enthusiasm for publishing this manuscript.
Introduction:

- The basic literature is not well written and does not even include any literature on alternative approaches with updated references regarding involvement of atypical antipsychotic treatment (aripiprazole, olanzapine or risperidone) in current observations.

- Authors fail to justify the correlation, and almost irrational and common information is present in the introduction part.

Material and methods:

-     Major drawback is the lack of supporting references and incomplete experimental and behavioral paradigms.

- Provide biochemicals kits numbers along with their city, country in all individual parameters.

- In order to support the assessment of all mentioned parameters in his study, the author should provide all the source documents and data he/she has followed for all assays and estimates.

- How was the dosing determined? Dose-responses should be performed.

- How was the sample size determined? Ideally, a priori sample size calculation should be performed to determine the appropriate sample size.
- Normality and variance homogeneity should be assessed across all groups of the same outcome variable and not individual experimental groups. If the data were not normally distributed or variance homogeneity was not met, nonparametric tests need to be performed.
Parametric data should be reported as mean +/- SD, while nonparametric data should be given/displayed as median and interquartile range. Longitudinal data should be analyzed using repeated measures tests.

Results:

-          Table 1,2, and 3 must be in graphical pattern. Convert all tables accordingly. Provide the statistical symbols accordingly.      

-          Results need more clarification and significant justification. Differentiating between the outcome and the discussion sections is quite difficult.

-          Use proper statistical reporting: i.e. for the results of each statistical test, the authors should report the statistical test that was applied, the test statistic (e.g. t, U, F, r), degrees of freedom as subscripts to the test statistic, and the exact probability value, including those for normality and variance homogeneity tests. Statistics should be reported in APA format, i.e.: t(df) = value, p = value; F(df1,df2) = value, p = value; r(df) = value, p = value; [chi]2 (df, N = value) = value, p = value; Z = value, p = value.  

-          Include statements on the tests for normality and variance heterogeneity and respective results. If the data were not normally distributed or variance heterogeneity was not met, nonparametric tests need to be applied.

Discussion:

-     To address the outcome of in-vivo measures/results separately and how they correlate with the existing literature, it would be better if the author restructured to take a more critical approach.
-     In both the discussion and the conclusion, the aims, rationale, and future perspectives are not evident clearly in relation with in-vitro experimentation.
-     The discussion is usually organized at the beginning to address all the observations and evaluate them at the end. It makes the results easier to contextualize and simpler to comprehend.

- Furthermore, a minimal critical analysis should be provided, along with current study limitations as well the future perspective as separate paragraph.

Conclusion:

-          Need to revise the conclusion in a scientific manner. Not accepted in its current form.

-          The authors should consider in future studies the neurotransmitter pathways involved in neuropathic rat behavior. Very small concentrations in the synaptic space of a discrete zone or nucleus may influence or induce a complex behaviour. The global concentration of the neurotransmitter in the whole brain may not reflect the specific behavioural pattern induced and the neural dynamics underlying it. The same could be said for peripheral determinations. They must be better justified.

-          This reviewer considers that this paper cannot be published in the present form. A detailed revision shortening, ordering and following the commented ideas could improve this interesting paper in a significant manner.

-          Several typewriting mistakes are present and needing correction. This reviewer remains at entire disposal for the next version.

Author Response

Revision 1

Reviewer 1

Dear Author,

Thanks for submitting your research manuscript entitled "Insulin-like growth factor 2 (IGF-2) and Insulin-like growth factor binding protein 7 (IGFBP-7) plasma protein levels are upregulated after atypical antipsychotic treatment in first episode drug-naïve schizophrenic patients ".

Before giving my final comments as well as the final revision of this manuscript, the author needs to address the following comments scientifically.

Dear reviewer,

Thank you very much for the detailed and precise review.

Changes made in the manuscript are indicated in red colour as marked by Word Microsoft Office.

Honestly, we appreciate the effort put in it and we hope to cover all the demands in order to improve the exposition of our work.

Major concerns: Please find out the following comments

  • The rationale and purpose behind selecting the Insulin-like growth factor (IGF) signaling system has been related with schizophrenia 16 (SZ) process are incomplete.
  • Title is misleading the reader. Title needs to reframed in simply manner accordingly.
  • The reviewer found irrational and non-scientific justification in the abstract—introduction and discussion part.
  • Abstract is very confusing. Irrational and fused with repetitions. Scientific output is not clear with this abstract.
  • The results and discussion are very poorly explained.
  • The reviewer feels the author needs to elaborate and justify it with proper citations and strong evidence. The author fails to explain the relevant justification in the introduction as mentioned in the discussion part.
  • A major drawback is a lack of supporting pre-clinical and clinical evidence regarding targeting drugs.
  • Complete mismatch of abstract, introduction, results and discussion.
  • Old and outdated references are not allowed. Try to update with relevant.

We have selected the IGF signalling system because there are plenty of studies (references 25-38) that are studying the link behind peripheral changes on these proteins and SZ symptomatology. There are several other studies that are studying the IGF signalling system in cognitive impairment and motor disability in age-related diseases such as neurodegenerative disorders (AD, Parkinson). We have not mentioned them because the scope of our study was to assess whether the levels of these proteins change as a result of long-term antipsychotic treatment or if it is a phenomenon that could be due to SZ itself.

We are in completely agree with the title and we have changed it for IGF-2 and IGFBP-7 are upregulated after atypical antipsychotics in schizophrenia patients to make emphasize in our main outcome because it is the one that shows that this increase is not due to long-term treatment conditioning.

We have also changed the abstract entirely, we hope that this issue is now better addressed and improved. We also improved the way we present statistical analysis in the results, by adding appropriate information. As we mentioned before we did not want to add pre-clinical data because this is a study based on human plasma and that would escape the scope of our purpose.

Conclusion has been entirely changed in order to make emphasise in the result of our tested hypothesis. We hope that now is better written.

We made the bibliography search in PubMed and all of them references that we include are still being referenced in present articles. However, in order to respect original work, we wanted to cite them.

We now specify each section. Again, thank you very much for the comments.

Title:

  • Mismatch of title with relevant introduction and conclusive remarks in the conclusion part.

As we mentioned before title has been changed. If further changes are needed, please let us know because we are open to suggestion. We selected the original title because we thought it was reflecting the main outcome of our work, but it might be long and confusing.

Abstract:

-     The rationale behind this research is not well explained, and several major concerns still constrain the reviewer's enthusiasm for publishing this manuscript.

The rationale is basically supported in past studies (28-37) that shows the IGF signalling system might act as a peripheral biomarker in SZ, and as we mention in the text there is still not clear what is the signalling or mechanisms that relate the IGF signalling system with psychiatric disorders.

Introduction:

- The basic literature is not well written and does not even include any literature on alternative approaches with updated references regarding involvement of atypical antipsychotic treatment (aripiprazole, olanzapine or risperidone) in current observations.

- Authors fail to justify the correlation, and almost irrational and common information is present in the introduction part.

Thank you for your appointments. We have written different manuscripts in an effort for improving the introduction part. Nevertheless, from our experience reading similar studies (28-37) this type of studies usually begin by explaining general aspects of SZ as a psychiatric disorder and then justify the rational of the study with past studies. From our perspective, it is just due to lack of information on this specific topic and that is why more studies are needed to fully approach. We estimate that we are just reaching the surface of the IGF signalling link not just with SZ but with other psychiatric conditions such as bipolar disorder and major depressive disorder.

Material and methods:

-     Major drawback is the lack of supporting references and incomplete experimental and behavioral paradigms.

- Provide biochemicals kits numbers along with their city, country in all individual parameters.

- In order to support the assessment of all mentioned parameters in his study, the author should provide all the source documents and data he/she has followed for all assays and estimates.

- How was the dosing determined? Dose-responses should be performed.

- How was the sample size determined? Ideally, a priori sample size calculation should be performed to determine the appropriate sample size.

- Normality and variance homogeneity should be assessed across all groups of the same outcome variable and not individual experimental groups. If the data were not normally distributed or variance homogeneity was not met, nonparametric tests need to be performed. Parametric data should be reported as mean +/- SD, while nonparametric data should be given/displayed as median and interquartile range. Longitudinal data should be analyzed using repeated measures tests.

We address the biochemical kits employed (not all is included in the main manuscript)

IGF-2: Human IGF-2 (Insulin-like growth factor 2) ELISA Kit (Fine Test ®, Catalogue Number EH0166. Wuhan Fine Biotech Co., Ltd. B9 Bld, High-Tech Medical Devices Park, No. 818 Gaoxin Ave. East Lake High-Tech Development Zone, Wuhan, Hubei, China(430206. Tel: (0086)027-87384275).

IGFBP-7: Human IGFBP-7 (Insulin-like growth factor-binding protein 7) ELISA Kit (Fine Test ®, Catalogue Number EH0171. Wuhan Fine Biotech Co., Ltd. B9 Bld, High-Tech Medical Devices Park, No. 818 Gaoxin Ave. East Lake High-Tech Development Zone, Wuhan, Hubei, China (430206. Tel: (0086)027-87384275).

We could not select any dosage for patients, this was a cross-sectional observational study. Even more, in chronic patients it is very complicate to elaborate dose-response analysis because their treatment conditions are frequently changed. This is worse in case of non-responders because normally they follow different treatments with different mechanisms of action which complicates any analysis.

Initially, we estimate a sample size of N = 100 per group (HC and SZ patients). However, sample size was constraint by many factors such as limited resources or unlikely events (such as covid-19 pandemic), since schizophrenia patients usually debut with first psychotic episodes in adolescence.

Moreover, the process of recruitment took us almost 3 years with an in-between pandemic which ultimately complicated the process. So, after discussing with our support team of statistics, we decided to reduce sample size and equalize it to the commonly found sample sizes in other studies of the same nature. For example: Akanji et al., (2007) recruited 53 vs 52 (ME vs HC), Yang et al., (2020) recruited 32 vs 30 (ME vs HC) and Chao et al., (2020) recruited 31 vs 30 (ME vs HC). In our case, we present a study with 40 vs 45 (ME vs HC) and 15 (FE) participants which should be considered as acceptable from our perspective in a study of these characteristic.

Whenever we made a comparison between groups the first analysis was a Shapiro-Wilk (S-W) test in SPSS (IBM SPSS Statistics 22) in order to check whether the distributions being compared were following or not a normal distribution (p > 0.05). We also look for skewness and kurtosis in case we have doubts if the distributions were following or not a normal distribution (normally, when p value were close to 0.05). We preferably used S-W and not Kolmogorov-Smirnov (K-S) due to sample size, since the second one is commonly recommended when N > 50.

If both distributions were normal then we use an independent parametric test to compare if there were statistical differences or not between means. In case one of the two distributions were not normal, then we selected a non-parametric test to compare medians. Since we like the style from Graph Pad Prism 7, we used this software to plot graphs and also to make statistical analysis. Now license numbers are appropriately indicated in the text.

In case of distributions from first episodes through time (FE vs FE_1) we only used the number of paired data. For example, in case of IGF-2 we only compared N = 9 and in case of IGFBP-7 we compared N = 11.

Results:

-          Table 1,2, and 3 must be in graphical pattern. Convert all tables accordingly. Provide the statistical symbols accordingly.     

-          Results need more clarification and significant justification. Differentiating between the outcome and the discussion sections is quite difficult.

-          Use proper statistical reporting: i.e. for the results of each statistical test, the authors should report the statistical test that was applied, the test statistic (e.g. t, U, F, r), degrees of freedom as subscripts to the test statistic, and the exact probability value, including those for normality and variance homogeneity tests. Statistics should be reported in APA format, i.e.: t(df) = value, p = value; F(df1,df2) = value, p = value; r(df) = value, p = value; [chi]2 (df, N = value) = value, p = value; Z = value, p = value. 

-          Include statements on the tests for normality and variance heterogeneity and respective results. If the data were not normally distributed or variance heterogeneity was not met, nonparametric tests need to be applied.

To elaborate the Tables, we have followed the template sheet that can be found at: https://www.mdpi.com/journal/ijms/instructions in section Submission Checklist, point 2. “Use the Microsoft Word template or LaTeX template to prepare your manuscript”. We do not know if this is correct or not, please let us if further changes are needed.

We have now completed the results section and we hope it is now better explained. We know that is difficult to differentiate between those sections because as we have mentioned before there are not too much data on this topic and more studies are needed to fully address this matter. We tried to discuss our results with a critical perspective but being cautious, since in this type of human studies there are a lot of parameters to take into account and we preferred to make clear and simple points.

We have now correct statistics and introduce APA style. We hope that these changes are enough, if not please let us know what changes might be introduced.

Discussion:

-     To address the outcome of in-vivo measures/results separately and how they correlate with the existing literature, it would be better if the author restructured to take a more critical approach.

-     In both the discussion and the conclusion, the aims, rationale, and future perspectives are not evident clearly in relation with in-vitro experimentation.

-     The discussion is usually organized at the beginning to address all the observations and evaluate them at the end. It makes the results easier to contextualize and simpler to comprehend.

- Furthermore, a minimal critical analysis should be provided, along with current study limitations as well the future perspective as separate paragraph.

We have now included a section termed Limitations and future perspectives (discussion) that we hope it covers the demand. Additionally, the conclusion has been entirely changed in order to make emphasize in what we think is the main outcome in our work.

We organized the discussion following the same order as the results, but we avoid to repeat titling on each topic because it is not indicated in the template sheet. A introductory paragraph might have been of help but we preferred not to include it because it would sound redundant with the following descriptions at the discussion section.

Conclusion:

-          Need to revise the conclusion in a scientific manner. Not accepted in its current form.

-          The authors should consider in future studies the neurotransmitter pathways involved in neuropathic rat behavior. Very small concentrations in the synaptic space of a discrete zone or nucleus may influence or induce a complex behaviour. The global concentration of the neurotransmitter in the whole brain may not reflect the specific behavioural pattern induced and the neural dynamics underlying it. The same could be said for peripheral determinations. They must be better justified.

-          This reviewer considers that this paper cannot be published in the present form. A detailed revision shortening, ordering and following the commented ideas could improve this interesting paper in a significant manner.

-          Several typewriting mistakes are present and needing correction. This reviewer remains at entire disposal for the next version.

We have now changed the conclusion entirely as we said before. We did not want to include any data from animal models in this article but we are working on a review that is trying to cover both animal models and in vitro experiments regarding the IGF signalling system in psychiatric disorders.

From our perspective, we wanted to be straightforward with our work and show how these peripheral proteins could be related with schizophrenia from a wide angle.

We have revised the whole text looking for typewriting mistakes and we hope that now they have been appropriately changed for the better.

Thank you so much for your comments, and we hope that all demands have been correctly addressed. If not, please let us know.

Sincerely yours,

Roberto Agís y Carlos Fernández

Reviewer 2 Report

The manuscript presented from Pereira et al., is interesting but not original. The authors reproduced similar experiment that have been also described in introduction. However they should motivated where is the novelty. At the same time they should use also qPCR strategy to quantify the transcripts. 

Author Response

Revision 1

Reviewer 2

The manuscript presented from Pereira et al., is interesting but not original. The authors reproduced similar experiment that have been also described in introduction. However, they should motivated where is the novelty. At the same time they should use also qPCR strategy to quantify the transcripts.

Dear reviewer,

First of all, we want to thank you for reviewing our work and for giving us a perspective we had not fully thought about until now.

Starting with the sentence “interesting but not original”, we want to specify that the innovative aspect introduced in our article is that is the first time that Insulin-like growth factor binding protein 7 (IGFBP-7) has been measured in a cohort of schizophrenic (SZ) Caucasian individuals, which is of interest because it allows the comparison with previous publications such as Yang et al., (2020) that was made in a Chinese population. Since ethnicity is considered a possible cofounding factor in some protein members of the Insulin-like growth factor (IGF) signalling system, this work will amplify the discussion and the establishment of new approaches.

We certainly “reproduced similar experiment” in a way, because the purpose of our investigation was also to measure peripheral protein levels. Following that, we used the ELISA technique. However, this was an interdisciplinary study and so we were able to add some interesting data obtained from blood analytics such as lipid serum values (total cholesterol and total triglyceride) and glucose blood levels to search for correlations with both proteins, as it was done separately in other works, but never altogether as we did. 

Finally, it is undeniable that the qPCR approach is of the utmost interest because it would expand our comprehension of the IGF signalling system in circulation. Nevertheless, we now present several reasons why we had not included it as a technique: A) We based our approach in past works where IGF-1 (references [25-35]) and IGF-2 ([30, 36-38]) plasma protein levels were measured in SZ. B) The main reason behind this choice is to find out whether these proteins could act as peripheral biomarkers in SZ and from our perspective it is the most suitable approach to introduce it in clinics in a realistic nearby future. C) On the other hand, transcripts in circulation are more volatile and susceptible to degradation and there are several processes that might affect the final product such as transcriptional or post translational modifications, etc D) The possible viewpoint would be measuring transcripts at brain level as it has been done in other works (such as in the case of IGFBP-4) to check correlations with peripheral levels, but this would require the obtention of post-mortem samples which clearly exceeds the scope of our study. Taking all of these points together we have finally decided not to measure transcripts levels, but we sincerely appreciate the observation and we will deeply study the possibility to add this technique in future research.

Sincerely yours,

Roberto Agís y Carlos Fernández

Reviewer 3 Report

Thank you for giving me the opportunity to review this manuscript.

This study found decreased IGF-2 plasma protein levels in a cohort of Caucasian FE drug-naïve patients when compared against controls. Moreover, this study indicated that both IGF-2 and IGFBP-7 significantly increased after atypical antipsychotic treatment. IGFBP-7 might be a potential biomarker of negative symptoms in first-episode psychosis in Schizophrenia.

I think it is necessary to revise the manuscript before publication.

1)     Please describe the study design more clearly. Was this a cross-sectional study? Please describe the setting, relevant dates, including periods of recruitment, exposure, follow-up, and data collection. Please define all predictors, potential confounders, and effect modifiers. Please describe any efforts to address potential sources of bias.

2)     Please explain how sample size was arrived at. I think that the sample size was small. Please describe any statistical methods to control for confounders. Please explain how missing data were addressed (Actually, the authors mentioned that 3 patients did not finish all the examinations). Please give unadjusted estimates and, if applicable, confounder-adjusted estimates and their precision. Please make clear which confounders were adjusted for and why they were included if possible.

3)     In the discussion section, the authors explained that ethnicity as a potential confounder, and the large-scale longitudinal studies are warranted. Please describe what kind of potential confounders should be adjusted in future studies, and how much the sample size should be in future studies.

I think it is necessary to revise the manuscript.

Author Response

Reviewer 3

Thank you for giving me the opportunity to review this manuscript.

Dear reviewer,

It is our pleasure that you could revise our work and make these interesting comments. We now proceed to answer point by point.

This study found decreased IGF-2 plasma protein levels in a cohort of Caucasian FE drug-naïve patients when compared against controls. Moreover, this study indicated that both IGF-2 and IGFBP-7 significantly increased after atypical antipsychotic treatment. IGFBP-7 might be a potential biomarker of negative symptoms in first-episode psychosis in Schizophrenia.

Indeed, this is the main outcome in our study since the actual literature (Yang et al., 2020) suggested to establish the difference between first and multiple episodes in order to check if antipsychotics are exerting this function or if it is something intrinsically related to the psychiatric disorder or if it is a consequence of an associated cognitive impairment. However, in order to fully prove this, more studies with bigger sample sizes are needed as we explain below.

I think it is necessary to revise the manuscript before publication.

1)           Please describe the study design more clearly. Was this a cross-sectional study? Please describe the setting, relevant dates, including periods of recruitment, exposure, follow-up, and data collection. Please define all predictors, potential confounders, and effect modifiers. Please describe any efforts to address potential sources of bias.

From our point of view, it could be partly considered as a cross-sectional study since we did not manipulate any variable. It was an observational study that provided information about what is happening with some members of the peripheral IGF system in a SZ Caucasian cohort. However, in case of the first episode (FE) group we have included more than one point in time (FE and FE_1) which means the study has also a longitudinal dimension in this group. Ultimately, this does not alter the outcome since the multiple episode (ME) group have been under treatment for a long period of time (years), which perfectly fit the approach in our study that is comparing the long-term effects that antipsychotics may exert on SZ patients.

We have now completed the Schizophrenic patients and healthy controls (Material and Methods) section with the solicitated data. If you consider that there is not enough information, please let us know.

Regarding the potential predictors and confounders, we have included a final section called Limitations and future perspective (Discussion) in which we state the main potential predictors/confounders. Briefly listed: a) the unknown correlation between brain and peripheral levels of these proteins, b) the source of the peripheral proteins could be surely traced to more than just one tissue (brain) of interest: liver, muscular tissue, etc. C) Proteins of the IGFs are believed to be dependent on some metabolic parameters such as body mass index (measured as BMI) and nutritional state (usually measured as serum albumin levels). We have now included the levels of serum albumin in the study in order to clear this point. D) We then repeat the importance of some genetic variables such as ethnicity and others related to the treatment itself (drug type, treatment regime and schedule, etc.) to emphasize the relevance of these aspects when designing a study of this nature.

2)           Please explain how sample size was arrived at. I think that the sample size was small. Please describe any statistical methods to control for confounders. Please explain how missing data were addressed (Actually, the authors mentioned that 3 patients did not finish all the examinations). Please give unadjusted estimates and, if applicable, confounder-adjusted estimates and their precision. Please make clear which confounders were adjusted for and why they were included if possible.

Initially, we estimated a sample size of N = 90 for each group (ME and healthy controls HC). In case of the FE group, we have to consider that these patients debuted with the psychiatric disorder which is not something very common (having in mind that the prevalence of SZ is around 1%) and in almost 3 years of recruitment we have only obtained N = 15 FE patients. Unluckily, the covid-19 pandemic difficulted our capacity to recruit more patients in the final course of the project.

From a statistical point of view, sample size might be considered small. Undoubtedly, we agree on that, and its why we suggest to make a study with a bigger sample size (lines 334 and 335), although we know the difficultness associated with the recruiting of FE drug-naïve patients. However, when we checked sample sizes from similar studies, we realized that they were almost the same (SZ vs HC): Akanji et al., (2007) 53 vs 52, Yang et al., (2020) 32 vs 30 and Chao et al., (2020) 31 vs 30. Something similar can be noticed in the IGF-1 studies.

Missing data was because those patients simply decided not to continue in the study, we are going to state that point clearer. Initially, we thought that age could be a potential confounder for IGF-2 as it happens with IGF-1 (which levels decrease with age as it has been widely proved). The same for gender (although this parameter is not that consistent showing differences in other studies). However, when we analysed IGF-2 values (and IGFBP-7 just in case) in an age and sex-dependant manner we did not obtain statistical differences in their levels (we now include that part of the analysis). We have also included the analysis for serum albumin (newly added) and the other parameters (lipids and glucose) just in case. This means that based on our analysis we have not found any confounding parameter that might be interfering in the final comparison.

On the other hand, one of the biggest limitations in our study is that we cannot discern between drug types for the FE group (aripiprazole, olanzapine and risperidone) since these patients received different dosage and combinations at different times, which was beyond our control since we were just making an observational study. Nonetheless, we expose Chao et al., (2020) work in which they found that IGF-2 levels increased regardless of the atypical antipsychotic administered. 

3)           In the discussion section, the authors explained that ethnicity as a potential confounder, and the large-scale longitudinal studies are warranted. Please describe what kind of potential confounders should be adjusted in future studies, and how much the sample size should be in future studies.

Indeed, as we stated above, we have decided to include a section named Limitations and future perspective (Discussion) in which we explore these aspects. We based our estimation not in a statistical strict approach, but in comparison to other studies that measure peripheral IGF-1 either in aging or in neurodegenerative disorders such as Parkinson’s disease where the focus is put on associated cognitive impairment and motor dysfunction. In these studies, sample sizes are normally huge. Okereke et al., (2006) (N = 376), Dik et al., (2002) (N = 1022), Al-Delaimy et al., (2009) (N = 1727), Pellechia et al., (2014) (N = 65 vs 60 HC), Jianfang et al., (2015) (N = 100 vs 76 HC), Fan et al., (2019) (N = 178 vs 23 HC), etc. However, we certainly know that in a case-control study is always a challenge to recruit such sample size.

On the other hand, we personally think that potential confounders are the ones listed in Limitations and future perspective (Discussion). Regarding limitations, special emphasise is put on treatment, despite the fact that drug type or dosage depended on clinician’s judgment which ultimate purpose is to guarantee the well-being of patients. Ideally, the same dosages or drug types should be administered. Nevertheless, our work is focus on humans and so this type of “limitation” is normally there and in our opinion, far from presenting an unreal scenery it gives a realistic panorama of these patients, who are commonly under different treatment regimes and drug types. Preclinical studies in mice models might be taken into account when considering the role of the IGF signalling system in psychiatric disorders and cognitive alterations.                             

I think it is necessary to revise the manuscript.

Thank you very much for revising our work and we hope that these points have been appropriately answered if not we are at your disposition for further suggestions.

Sincerely yours,

Roberto Agís y Carlos Fernández

Round 2

Reviewer 1 Report

Dear Author, 

After careful consideration, reviewer feels that revised manuscript improved much. And can be accepted in current form. 

Author Response

Revision Round 2

Reviewer 1

Dear Author, 

After careful consideration, reviewer feels that revised manuscript improved much. And can be accepted in current form. 

Dear reviewer,

Thank you very much for your appreciations.

Following your indications, we have been able to improve the exposition of our work.

Sincerely yours,

Roberto Agís & Carlos Fernández

Reviewer 2 Report

The authors have not satisfied my concerns, I continue to think that this article is not original and they should quantify the transcripts

Author Response

Revision Round 2

Reviewer 2

The authors have not satisfied my concerns, I continue to think that this article is not original and they should quantify the transcripts

Dear reviewer,

We sincerely appreciate the comments, we now proceed to give an explanation about the originality of our work and why we did not measure the transcripts.

In 1996, Hopkins et al., measured that both circulating IGFBP-3 and IGF-2 values were significantly higher in Caucasians than in an Asian population (N = 53). In recent years, several publications [25-35] have measured circulating levels from members of the IGF signalling system in schizophrenia (SZ) chronic and debutant patients and searched for correlations with SZ symptomatology (PANSS). Recently, Yang et al., (2020) found lower levels of IGF-2 and IGFBP-7 in chronic patients that had been without treatment for 3 months before protein measurement. However, the situation in first episodes could be different and would help to understand if these changes are intrinsically related to SZ itself or if they resulted from a long-term antipsychotic treatment. So, measuring these proteins in SZ drug-naïve debutant patients (first episode) could be of interesting in order to distinguish their effects from chronic patients (even when they were without treatment). As we mentioned before, ethnicity may be interfering with the conclusion of this outcome, since the work from Yang et al., (2020) was made in an Asian population and so the interpretation may vary between populations. Having in mind these aspects, our work should be rightfully considered as original because: it is made in a Spanish (Caucasian) population and more importantly in first episode schizophrenic patients. Moreover, is the first time that circulating IGFBP-7 levels have been measured in SZ debutant patients. For these main two reasons we certainly consider that our work is an original source.

Despite the fact that we have gave our reasons why transcripts have not been measured in the last report, after reading them again we thought another question regarding previous studies that might be an interesting approach to explore. Our major concern is “why IGF transcripts have not been previously measured in those works?” [references 25-35]. From our point of view, there might be a logical reason why transcripts levels were not addressed. Actually, we do not possess a definitive answer, but after some queries we have definitively widen our perspective as follow [consider all the points mentioned in the last revision: Finally, it is undeniable that the qPCR approach is of the utmost interest because it would expand our comprehension of the IGF signalling system in circulation. Nevertheless, we now present several reasons why we had not included it as a technique: A) We based our approach in past works where IGF-1 (references [25-35]) and IGF-2 ([30, 36-38]) plasma protein levels were measured in SZ. B) The main reason behind this choice is to find out whether these proteins could act as peripheral biomarkers in SZ and from our perspective it is the most suitable approach to introduce it in clinics in a realistic nearby future. C) On the other hand, transcripts in circulation are more volatile and susceptible to degradation and there are several processes that might affect the final product such as transcriptional or post translational modifications, etc]. To the best of our knowledge, there is only one study that was made by Fromer et al., (2016) that proved that IGF-2 is the top downregulated gene in the prefrontal cortex of schizophrenic patients. From our perspective, mRNA measurements (in a context of psychiatric disorder such as schizophrenia) would only fit if they were done in post mortem brain samples as other works have done (i.e., mRNA IGFBP-4 that has been recently found downregulated in hippocampal brain sections of SZ patients). What we are trying to address in our study is the possible biomarker role of these proteins at a peripheral level because is the outcome that has been dawning much attention in this field during the last years. We were not interested on measuring transcripts of these proteins as past works were not (25-35) because it would not reflect brain conditions and may mislead the focus of the study which could be finally translated as a misuse of resources. As we said before, we just wanted to complement recent studies in this field proving the main hypothesis that is “whether protein peripheral levels were altered in SZ by long-term treatment conditioning or if it is something related to SZ itself”. In our sincere opinion, this is the question that others researchers were trying to answer and that is why they might have not measured transcripts too. 

We hope that your concerns have been finally satisfied. We also want to thank you for reviewing our work and enable us to think about transcriptomics which we think it would constitute a novel approach in this specific scenery (IGF circulating levels in SZ).

Sincerely yours,

Roberto Agís & Carlos Fernández

Reviewer 3 Report

Thank you for giving me the opportunity to review this manuscript.

I think it is still necessary to revise the manuscript because of the following minor points.

1) I think it is important to describe that this study is, for example, "the results of a stopped cross-sectional study in Spain" in the title and the abstract.

2) Please describe any sample size estimation plan in the method section. Please describe why sample size did not arrive at the estimated one in the discussion section. 

For example, "We estimated that sample size was 90 in each group, but the number of participants was much smaller than the estimated one. That is partly because the COVID-19 pandemic made it difficult to recrult participants."

Because you explained that "the actual literature (Yang et al., 2020) suggested to establish the difference between first and multiple episodes in order to check if antipsychotics are exerting this function or if it is something intrinsically related to the psychiatric disorder or if it is a consequence of an associated cognitive impairment. However, in order to fully prove this, more studies with bigger sample sizes are needed as we explain below." Furthermore, you tried to prove this in this study. Actually the estimated sample size was 90 for each group. This sampling strategy is not statistic but descriptive.  However, the authors cannot do it due to the COVID-19 pandemic.

3) Please delete the sentences of  "To the best of our knowledge, this is the first time that IGFBP-7 plasma protein levels have been evaluated in a cohort of Caucasian FE drug-naïve SZ group of patients." to avoid misleadings. Instead, please describe that this study was planned to be a empirical study. Furthermore, the novelty of this study might be that IGFBP-7 plasma protein levels have been evaluated in a cohort of  FE drug-naïve SZ group of patients in Spain.

I am sorry to inform you that it is still necessary to revise the manuscript due to the minor points.

Author Response

Revision Round 2

Reviewer 3

Thank you for giving me the opportunity to review this manuscript.

I think it is still necessary to revise the manuscript because of the following minor points.

Dear reviewer,

Thank you very much for your comments. Curiously, some of your concerns have been previously discussed between members of our group.

We now proceed to answer them. We hope to cover all the demands.

1) I think it is important to describe that this study is, for example, "the results of a stopped cross-sectional study in Spain" in the title and the abstract.

We used G*Power 3.1.9.7 Software to estimate total sample size per group. We select F test family and ANOVA as statistical test. We estimate an effect size of 0.5, significance of 95% (α = 0.05), and potency of 90% (β = 0.01). The number of groups were fixed at 3 (HC, FE and ME). As a result, we obtained a N = 91 per group. However, this estimation has to meet two main limitations. First, in case of FE we are using incidence instead of prevalence (as we use for ME patients) what could be simply translating into “is considerably less likely to find FE drug-naïve patients than ME patients”, because the occurrence of a first episode in adulthood is not that common as it is during adolescence. Second, covid-19 pandemic difficulted the recruitment process because blood extraction occurred when these patients were hospitalized as it is indicated in the manuscript.   

Initially, it was our intention to put it as “interrupted” (which can be considered as a synonym for “stopped”). However, after discussing this issue we came to the conclusion that in fact, our final sample size (not 90 per group, but N = 15 for FE, N = 40 for ME and N = 45 for HC) was similar (even bigger than) to previous studies [30, 36 & 37] made in this specific field (circulating IGFs in SZ).

From our sincere perspective, including “stopped” in a sentence like “the results of a stopped cross-sectional study in Spain” would not add something that is really meaningful for the research, although it was part of the process.

2) Please describe any sample size estimation plan in the method section. Please describe why sample size did not arrive at the estimated one in the discussion section.

For example, "We estimated that sample size was 90 in each group, but the number of participants was much smaller than the estimated one. That is partly because the COVID-19 pandemic made it difficult to recrult participants."

Because you explained that "the actual literature (Yang et al., 2020) suggested to establish the difference between first and multiple episodes in order to check if antipsychotics are exerting this function or if it is something intrinsically related to the psychiatric disorder or if it is a consequence of an associated cognitive impairment. However, in order to fully prove this, more studies with bigger sample sizes are needed as we explain below." Furthermore, you tried to prove this in this study. Actually, the estimated sample size was 90 for each group. This sampling strategy is not statistic but descriptive.  However, the authors cannot do it due to the COVID-19 pandemic.

We have introduced this change in the Statistical Analysis (Material and Methods) (indicated in the text with tracking mode).

Thank you very much for this appreciation.

Initially, we did not want to include this type of message because we thought that (although estimation was not met) sample size was similar to other references at the end. However, now it might be clear and that will definitively improve the exposition of our work.

3) Please delete the sentences of "To the best of our knowledge, this is the first time that IGFBP-7 plasma protein levels have been evaluated in a cohort of Caucasian FE drug-naïve SZ group of patients." to avoid misleadings. Instead, please describe that this study was planned to be a empirical study. Furthermore, the novelty of this study might be that IGFBP-7 plasma protein levels have been evaluated in a cohort of FE drug-naïve SZ group of patients in Spain.

Absolutely, we have changed that sentence.

Just for mentioning.

We just wanted to write it as “to the best of our knowledge” in order to make a clear emphasis that we did not find any previous reference that had measured circulating IGFBP-7 levels in SZ Spanish patients.

We put Caucasian instead of Spanish (or something similar) because Spanish people could be considered as Caucasians, since Hopkins et al., (1996) evaluated IGF-2 between different ethnicities in Caucasians (among other ethnicities) and the Spanish people would only fit in that one. Nevertheless, we agree that a term like “Caucasian” might be misleading and confusing in a less wide comparison.

I am sorry to inform you that it is still necessary to revise the manuscript due to the minor points.

Thank you very much for your comments,

We have made the changes as you have suggested and everything is marked in the manuscript in the tracking mode.

Sincerely yours,

Roberto Agís & Carlos Fernández

Round 3

Reviewer 2 Report

The authors satisfied my major concern motivating the choice to not use qPCR approach. I propose the acceptance in the present form.

Author Response

Revision Round 3

Reviewer 2

The authors satisfied my major concern motivating the choice to not use qPCR approach. I propose the acceptance in the present form.

Thank you very much for your comments throughout this revision process.

You have opened our perspective on this matter in which we have been working for months and it is the first time that we have asked ourselves such an important question.

Sincerely yours,

Roberto Agís & Carlos Fernández

Reviewer 3 Report

Thank you for giving me the opportunity to review the revised manuscript.

I think this manuscript would be suitable for publication after a minor revision. Please add "Spanish" in the title.

In other words, please change the title as " Insulin-like growth factor 2 (IGF-2) and Insulin-like growth 3 factor binding protein 7 (IGFBP-7) are upregulated after 4 atypical antipsychotic in Spanish schizophrenia patients"

Thank you for your hard work on revising this manuscript.

Author Response

Revision Round 3

Reviewer 3

Thank you for giving me the opportunity to review the revised manuscript.

I think this manuscript would be suitable for publication after a minor revision. Please add "Spanish" in the title.

In other words, please change the title as " Insulin-like growth factor 2 (IGF-2) and Insulin-like growth 3 factor binding protein 7 (IGFBP-7) are upregulated after 4 atypical antipsychotic in Spanish schizophrenia patients"

Title has been changed according to your suggestion. Actually, we think it helps to rapidly understand one of the main points in our study (and the present discussion among IGFs and SZ) that is the ethnicity of the patients.

Thank you for your hard work on revising this manuscript.

Thank you for this comment.

We are very grateful for your careful revision because it has helped us to improve the exposition of our work and we hope to continue on this line on future research.

Sincerely yours,

Roberto Agís & Carlos Fernández

This manuscript is a resubmission of an earlier submission. The following is a list of the peer review reports and author responses from that submission.